

# A comprehensive global oceanic dataset of helium isotope and tritium measurements

William J. Jenkins[1], Scott C. Doney[1,2], Michaela Fendrock[1], Rana Fine[3], Toshitaka Gamo[4], Philippe Jean-Baptiste[5], Robert Key[6], Birgit Klein[7], John E. Lupton[8], Monika Rhein[9], Wolfgang Roether[9], Yuji Sano[4], Reiner Schlitzer[10], Peter Schlosser[11], Jim Swift[12]

*Correspondence to*: William J. Jenkins (wjenkins@whoi.edu)

**Abstract.** Tritium and helium isotope data provide key information on ocean circulation, ventilation, and mixing, as well as the rates of biogeochemical processes, and deep-ocean hydrothermal processes. We present here global oceanic datasets of tritium and helium isotope measurements made by numerous researchers and laboratories over a period exceeding 60 years. The dataset has a DOI:10.25921/c1sn-9631 and is available at https://www.nodc.noaa.gov/ocads/data/0176626.xml, and includes approximately 60,000 valid tritium measurements, 63,000 valid helium isotope determinations, 57,000 dissolved helium concentrations, and 34,000 dissolved neon concentrations. Some quality control has been applied in that questionable data have been flagged and clearly compromised data excluded entirely. Appropriate metadata has been included, including geographic location, date, and sample depth When available, we include water temperature, salinity, and dissolved oxygen. Data quality flags and data originator information (including methodology) are also included. This paper provides an introduction to the dataset along with some discussion of its broader qualities and graphics.

## 1 Introduction

The global oceanic distributions of tritium ($^3$H, a radioactive isotope of hydrogen with a 12.3 years), its daughter product $^3$He, and helium isotopes in general arise from the complicated interplay of ocean ventilation, circulation, and mixing, with

---

[1] Woods Hole Oceanographic Institution, Woods Hole, MA 02543, USA
[2] Now at University of Virginia, Charlottesville, VA 22904, USA
[3] RSMAS, University of Miami, Miami. FL 33149, USA
[4] Atmosphere and Ocean Research Institute, The University of Tokyo, Kashiwa, Chiba 277-8564, Japan
[5] LSCE, CEA-CNRS-UVSQ, CEA/Saclay, 91191 Gif-sur-Yvette cedex, France
[6] Princeton University, Princeton, New Jersey, USA
[7] BSH, 20359 Hamburg, Germany
[8] NOAA Pacific Marine Environmental Laboratory, Newport, Oregon, USA.
[9] IUP, University Bremen, D28359 Bremen, Germany
[10] Alfred Wegener Institute, 27568 Bremerhave, Germany
[11] LDEO, Columbia University, Palisades, NY, USA
[12] SIO, University of California San Diego, La Jolla, CA, USA



the hydrologic cycle, air-sea exchange, and geological volatile input. Observations of the delivery of tritium to the ocean and its redistribution are a useful tool for diagnosing gyre- and basin-scale ventilation and circulation (Doney et al., 1992; Doney and Jenkins, 1994; Dorsey and Peterson, 1976; Dreisigacker and Roether, 1978; Fine and Ostlund, 1977; Fine et al., 1987; Fine et al., 1981; Jenkins et al., 1983; Jenkins and Rhines, 1980; Michel and Suess, 1975; Miyake et al., 1975; Ostlund,

1982; Sarmiento, 1983; Weiss and Roether, 1980; Weiss et al., 1979).

In shallow waters, away from sea-floor hydrothermal vents, the combination of tritium and $^3$He may be used to determine the time elapsed since a water parcel was at the sea surface, making it a useful tool for diagnosing ventilation and circulation on seasonal through decade time-scales (Jenkins, 1987, 1998, 1977). The ingrowth and evasion of tritiugenic $^3$He from the thermocline is also useful as a flux gauge for constraining the rate of nutrient return to the ocean surface (Jenkins, 1988;

Jenkins and Doney, 2003; Stanley et al., 2015) as well as upwelling in coastal regions (Rhein et al., 2010). Finally, the distribution of helium isotopes in the deep sea provides important quantitative constraints on the impact of submarine hydrothermal venting on many elements because the global hydrothermal helium flux is well known (Bianchi et al., 2010; Holzer et al., 2017; Schlitzer, 2016). This makes $^3$He useful as a flux gauge (German et al., 2016; Jenkins et al., 1978; Jenkins et al., 2017; Lupton and Jenkins, 2017; Resing et al., 2015; Roshan et al., 2016). Consequently, there have been

numerous measurements of these properties over the years, particularly under the aegis of major observational programs like GEOSECS, TTO, WOCE, CLIVAR, GO-SHIP, and GEOTRACES. It seems valuable to assemble all existing data, including those measured prior to and outside of these programs, along with appropriate metadata, in one place to facilitate further use and analysis. This is a report of these efforts.

## 2 Methods

### 2.1 Tritium measurement methodology

There are at present three distinct methods for the determination of tritium in water samples: direct measurement of tritium abundance by accelerator mass spectrometry (AMS), radioactive counting of tritium decay rate, and daughter product ($^3$He) ingrowth method. The first method (AMS) has not been used for the measurement of environmental tritium levels, but is better suited to measurement high tritium concentrations in small samples, largely for biomedical tracer research (Brown et

al., 2005; Chiarappa-Zucca et al., 2002; Glagola et al., 1984; Roberts et al., 2000). The second method usually involves enrichment of the water samples either by electrolysis (e.g., Ostlund and Werner, 1962) or thermal diffusion (Israel, 1962) followed by low level radioactive counting; either by liquid scintillation (Momoshima et al., 1983) or gas proportional counting (Bainbridge et al., 1961). Measurements are made relative to prepared standards (Unterweger et al., 1980) and accuracy appears to be limited by the reproducibility of the enrichment process to 3-10% (Cameron, 1967).

The third method, $^3$He ingrowth, is a three step method. First, it involves degassing of a quantity (~1 to 1000 ml) of water to remove all dissolved helium. Second, the degassed water is stored in a helium leak-tight container (usually low He-permeability aluminosilicate glass or metal) for a period of several weeks to a year or more. Experience indicates that it is





necessary to shelter the stored samples from cosmic rays since there is a latitude-dependent cosmogenic $^3$He production rate that masquerades as "tritium signal" (Lott and Jenkins, 1998). Finally, the ingrown $^3$He is extracted from the water sample and mass spectrometrically analysed (Clarke et al., 1976; Ludin et al., 1997). The last method, although it involves a potentially lengthy incubation period, is chemically simpler and does not involve isotopic enrichment steps. As such, it offers

intrinsically greater accuracy (limited by standardization of the mass spectrometer, typically better than 1%) and lower ultimate detection limit (Jenkins et al., 1983; Lott and Jenkins, 1998).

## 2.2 Helium isotope measurement methodology

Water samples are usually drawn from Niskin bottles into a helium leak-tight container either for ship-board (Lott and Jenkins, 1998; Roether et al., 2013) or shore-based gas extraction. The latter involves either clamped (Weiss, 1968) or

crimped copper tubing (Young and Lupton, 1983). The extracted gases are subsequently purified and concentrated, usually cryogenically (Lott, 2001; Lott and Jenkins, 1984; Ludin et al., 1997) and expanded into a mass spectrometer for isotopic analysis. While time-of-flight mass spectrometry has been used (Mamyrin, 2001; Mamyrin et al., 1970) most oceanic helium isotope measurements have been made using specially designed magnetic sector instruments (Bayer et al., 1989; Clarke et al., 1969; Lott and Jenkins, 1998, 1984; Ludin et al., 1997). Measurements are typically standardized to marine air and

corrected for any sample-size-dependent ratio effects determined by measurement of different-sized air aliquots. Depending on the amount of tritium in the water and the length of time a water sample is stored prior to gas extraction, it is generally necessary to correct the $^3$He/$^4$He results for decay of tritium during storage. For very deep and old samples, where tritium concentrations are very low, this correction may be inconsequential.

## 20 2.3 Helium and neon concentration measurements

Helium and neon concentration measurements are typically made by mass spectrometric peak height manometry; that is, by comparison of major isotope ion currents ($^4$He and $^{20}$Ne) between the unknown and helium/neon derived from an aliquot of marine air. The air aliquot size is determined from a knowledge of the barometric pressure, relative humidity, and

temperature at which the previously evacuated air standard reservoir was filled, and the volume of the aliquot. Generally the ion current is assumed to be a linear function of the sample size (number of atoms) over some narrow range, but also can be corrected by construction of a standard curve using multiple aliquots. Some measurements (notably the GEOSECS expedition) were made by splitting the extracted water sample and measuring the helium and neon contents using isotope dilution (with $^3$He and $^{22}$Ne spikes).

## 30 2.4 Data organization

We have compiled a comprehensive dataset consisting of helium isotope and tritium measurements in oceanic waters made by numerous laboratories over the past 6 decades. The dataset includes ~60,000 tritium and ~63,000 helium isotope



measurements, ~57,000 dissolved helium concentrations, and ~34,000 dissolved neon concentrations in ocean water taken from 1952 to 2015 (for tritium) and from 1967 to 2015 for helium. In additional to "spot sampling", there are ~ 380 cruises, with sampling from >5,400 locations for tritium and ~5,600 locations for helium. The helium data are from 8 different laboratories, and the tritium data from 15 laboratories world-wide. In addition to including measurement uncertainties, a data

quality flag, and data source, each data point is accompanied by location (latitude, longitude, depth) and time (decimal year) of sampling. When available, water temperature, salinity, and dissolved oxygen measurements are included.

A number of the earliest measurements were obtained from publications. In those cases, the publication source is given. If the data were transcribed from tables, the table number and page is also given. In the event that the data were only available graphically, a computer program to digitize the data from plots was used, and in the rare cases where graph quality was

10 sufficiently poor to degrade the precision of the data, the uncertainties were commensurately increased to reflect it. Where data had been assigned a Digital Object Identifier (DOI), this is also included.

The dataset consists of three tables. The REFERENCES table is a list of the data sources keyed by the text variable "Reference_Code" found in the main data table. This should in principle provide attribution and/or more information

regarding the data origin. The METHODS table provides a more complete description of the methods fields "Tritium_Method" and "Helium_Method" in the main data table. This is intended to provide useful interpretive information regarding how the sampling and/or measurements were accomplished. The main data table fields are described in the Table 1. Most data fields have an associated quality flag field whose meaning is summarized in Table 2. Following WOCE ocean data convention, normal acceptable data are associated with a quality flag of 2, whereas questionable data have a flag of 3.

Results obtained by averaging 2 or more replicates are signified with a flag of 6. When fields are missing for a given record, the data is entered as -999 and the corresponding quality flag is 9. The tritium, helium, helium isotope, and neon data also have an associated uncertainty field (e.g., "Tritium_Error") which is the estimated uncertainty in the data points. This is either provided by the data measurer or an estimate based on described procedures, and can vary greatly between methods and laboratories so the user is advised to be aware of this value.

In the spirit of the WOCE/CLIVAR/GO-SHIP[13] convention, the combination of ExpoCode, Station, CastNo, and Bottle should uniquely define a sample. That is, no two data records should have the same combination of these values. This has been followed with most of the information here: when a sample's station, cast, or bottle number were not provided (in the case of literature data), arbitrary but unique numbers were assigned. In order to supplement this identification we added a

30 unique integer record ID number.

---

[13] WOCE is the World Ocean Circulation Experiment (e.g., see https://www.nodc.noaa.gov/woce/ ), CLIVAR is the Climate and Ocean – Variability, Predictability, and Change (e.g., see http://www.clivar.org/about), and GO-SHIP is the Global Ocean Ship-Based Hydrographic Investigations Program (see http://www.go-ship.org/ )



## 2.5 Data formats and availability

The data set's DOI is 10.25921/c1sn-9631. The data is available for download at the U.S. National Oceanic and Atmospheric Administration's National Centers for Environmental Information web site https://www.nodc.noaa.gov/ocads/data/0176626.xml in a number of formats for maximum flexibility. For maximum

flexibility, we suggest one of the following three database formats: Microsoft Access®, PostgreSQL, or ODV (Ocean Data View). In addition, the three tables are available as 4 files (the main data table is split in two to avoid spreadsheet row-number limitations) in Microsoft Excel® or as a comma separated plain text files. Finally, the data table is available for download as a MATLAB® binary data file.

## 3 Graphics and examples

We provide some example graphics to indicate the scope and nature of the data holdings. These include time histories of analyses per year for both types of measurements (Figure 1) and maps of sampling locations (Figure 2). The intent is to provide a broad overview of the character of the data sets while not over-interpreting its details and features.

The temporal distribution of oceanic tritium measurements begins in the early 1950s with the development of enrichment and counting capabilities suitable for environmental levels, and the recognition of the existence of cosmogenic tritium

production (Cornog and Libby, 1941; Currie et al., 1956; Grosse et al., 1951; Kaufman and Libby, 1954; Libby, 1946) and the desire to measure its distribution in the hydrologic cycle. The advent of atmospheric thermonuclear tests in the 1950s and early 1960s dwarfed the natural global inventory (Weiss and Roether, 1980), which motivated an increase in oceanic measurements in the 1960s. The initiation of global ocean chemistry, hydrographic, and tracer survey efforts (especially GEOSECS[14]) further increased this activity. A final boost to tritium measurement rates occurred with the development of the

$^3$He regrowth method (Clarke et al., 1976) coupled with even more ambitious global surveys (such as WOCE, CLIVAR and GO-SHIP).

The helium sampling time history was basically initiated and motivated by the discovery of primordial $^3$He injection into the deep waters (Clarke et al., 1969) which drove the inclusion of helium isotope measurements in the GEOSECS program. It was quickly realized that the existence of *tritiugenic* $^3$He (that produced by the *in situ* decay of tritium) offered the potential

for a dating tool as well (Jenkins et al., 1972; Jenkins and Clarke, 1976), which spurred continued helium isotope measurements in the global surveys. These were enabled by a number of laboratories coming "on-line" in the 1970s and 1980s.

The tritium and helium sampling locations shown in Figure 2 are dominated by the global survey programs cruise tracks, but also include a number of ocean island monitoring sites (especially for tritium). One difference in the two maps is the extra

---

[14] GEOSECS is the Geochemical Ocean Sections Survey





tritium sampling in the Arctic, perhaps largely driven by H.G. Ostlund's motivation to exploit this isotope's potential in studying the Arctic fresh water system (Ostlund, 1982).

It is well known that the delivery of bomb tritium to the ocean was a reflection of the distribution of the atmospheric tests, and occurred in two principle modes: a dominant pulse-like injection in the northern hemisphere and a much smaller more

"diffuse" input into the southern hemisphere (Doney et al., 1992). This can be seen in Figure 3, which is plot of near surface (<50 m depth) water tritium concentrations vs. time for number of latitude bands. Most striking is the northward increase in the concentration (y-axis) ranges. The time-latitude trends in Figure 3 reflect this globally asymmetric delivery, but much of the structure caused by regional variations in atmospheric input and ocean circulation may be masked due to conflating major ocean basins in the groupings.

As a transient tracer, tritium offers an opportunity to visualize the ventilation of deep waters on decade to century time scales. A bench-mark observations of water column tritium concentrations during the GEOSECS Atlantic Expedition reveals North Atlantic deep water formation in a graphic manner (Ostlund et al., 1974). Valuable also is the evolution of the tritium distributions as they penetrate the subtropical thermocline and also intermediate and deep waters. Figure 5 shows four "snapshots" of tritium distributions along a section along approximately 52°W in the North Atlantic between the South

American and North American coasts (see map inset). Tritium concentrations are decay-corrected to a common mid-point in time (1997) for comparison. While the last three occupations are conveniently along the same cruise track (courtesy WOCE/CLIVAR/GO-SHIP), the first is a composite from several cruises of opportunity taken over an approximately 1 year period at roughly the same longitude. One can readily see the downward propagation and ultimate dispersion of the bomb tritium pulse within the main thermocline (the upper 1000 m) and the progressive ingrowth of tritium at intermediate (1500-

2500m depth) and in the bottom layers (~4000 m) is striking. Equally important is the bottom contour-hugging nature of the deep and intermediate levels, with a delayed arrival in the south corresponding to the southward propagation of the transient tracer along the deep western boundary current system from Nordic and Labrador Seas (Doney and Jenkins, 1994) .

Figure 4 shows the corresponding helium isotope anomaly distribution for those sections. Interpretation is a little more complicated, but the build-up of *tritiugenic* $^3$He within the main thermocline is an important diagnostic of vertical transport

for the subtropical main thermocline. Its retention and back-flux to the ocean surface is a uniquely valuable transient tracer observation, one that parallels the build-up and reflux of inorganic nutrients in the thermocline (e.g., nitrate and phosphate) but in a quantifiably defined manner. Observations of surface water $^3$He excesses (not shown here) have been used as flux gauges to quantify/constrain regional-scale new production rates (Jenkins, 1988; Jenkins and Doney, 2003; Stanley et al., 2015) as well as upwelling rates (Rhein et al., 2010).

We also include a time series (plotting only the upper 2000 dbar) for stations within 200km of Bermuda in the subtropical North Atlantic, which highlights some high temporal resolution features in the penetration of tritium into, and ingrowth of $^3$He within the main thermocline and intermediate waters at one location.



Perhaps the most notable features of the oceanic distribution of excess $^3$He are the large tongues emanating from mid-ocean ridges and other volcanic edifices on the sea floor (see Figure 7), driven by the roughly order-of-magnitude higher $^3$He/$^4$He ratio in the earth's mantle compared to the atmosphere (Clarke et al., 1969; Kurz and Jenkins, 1981; Kurz et al., 1982; Lupton and Craig, 1975). It was in fact the initial discovery (Clarke et al., 1969) that sparked continued interest in the

oceanic distribution of helium isotopes. The ongoing survey of deep helium features on both large and small scales continues today, particularly in support of "flux gauge" studies of other trace elements and metals influenced by seafloor hydrothermal processes (e.g., Jenkins et al., 2017; Resing et al., 2015; Roshan et al., 2016). These are based on recent model-based estimates of the global flux of hydrothermal $^3$He, which center around 550 mol/y (Bianchi et al., 2010; Holzer et al., 2017; Schlitzer, 2016). This estimate can be usefully compared to the expected global flux of *tritiugenic* $^3$He. The global tritium

production by atmospheric nuclear weapons tests has been estimated to be of order 3 GCi (Weiss and Roether, 1980, corrected from 1972 to 1963) to 5 GCi (Michel, 1976). Given that the bulk of the tritium "impulse" entered the hydrologic system, and subsequently the oceans within a decade or so, one can argue that the production rate of tritiugenic $^3$He was of order 3000 mol/y in the mid 1970s. By the mid 1990s, this would be of order 1000 mol/y. Separating the two "types" of $^3$He (tritiugenic *vs.* primordial) in the northern hemisphere, where the bulk of the tritium delivery occurred (Doney et al., 1992;

Weiss and Roether, 1980), is relatively simple in that the former appears largely at the sea surface and the latter is concentrated in older, deeper waters. The separation in the southern hemisphere is not so simple.

**4 Acknowledgments**

This dataset represents the hard work over many decades of numerous individuals that are not included in the authorship list of this paper. We list their names and affiliations at the time of their contributions in Table 3. The list focuses on those who

made the measurements rather than those who may have used the data. We apologize if there are others that we may have missed in this list.

We also would like to recognize that the ability to make the measurements presented in this dataset was a consequence of the pioneering work of more than a few inventive and talented individuals. While space does not permit mentioning them all here, we felt it appropriate to highlight a pair of pioneering scientists who conducted landmark studies on ocean tritium and

$^3$He measurements.

**4.1 W. Brian Clarke (1937-2002)**

Although not the first to measure $^3$He/$^4$He in the environment (that was done by Aldrich and Nier, 1948), Brian Clarke made the first reported helium isotope measurements in seawater (Clarke et al., 1969). He made his first measurements using a modified single stage magnetic sector, single collector mass spectrometer to a precision of about 2%. Brian developed the

first compact all-metal branch tube mass spectrometer specifically designed to make $^3$He/$^4$He measurements ultimately to a precision of 0.1 to 0.2%. At the time, conventional wisdom dictated that such measurements (let alone *precision*





measurements) were not possible with a single stage magnetic sector instrument for such high ($10^6$) abundance ratios, but Clarke forged ahead anyway. He initially constructed two instruments in the early 1970s, using one at McMaster University in Hamilton, Ontario, Canada, and setting the other machine up at the Scripps Institute of Oceanography in La Jolla, CA, USA for H. Craig and J. Lupton. These instruments were used in support of the GEOSECS program, and subsequently for a

wide variety of other research projects. One of his students (Jenkins) moved to the Woods Hole Oceanographic Institution in Woods Hole, MA, USA, where he extended Clarke's design to construct three other instruments. A post-doctoral investigator from his laboratory (Z. Top) moved to RSMAS at the University of Miami and constructed a similar machine. Brian also collaborated with a U.K. mass spectrometer company to make a commercially produced mass spectrometer available to the global scientific community.

In his early career, Brian contributed to the study of meteorites and nuclear physics using isotope mass spectrometry. Beginning in 1969 Clarke published a series of ground-breaking papers on $^3He/^4He$ measurements in seawater and lakes (Clarke et al., 1969, 1970; Clarke and Kugler, 1973; Craig and Clarke, 1970; Craig et al., 1975; Jenkins and Clarke, 1976; Top and Clarke, 1983; Torgersen and Clarke, 1985). He contributed research to geology, hydrology, limnology, nuclear physics, medicine, and other disciplines, and was active until 2002 publishing on $^3He$-related evidence for/against cold

fusion (Clarke, 2001; Clarke and Oliver, 2003; Clarke et al., 2001).

In addition to developing a mass spectrometer capable of measuring $^3He/^4He$ ratios to order 0.1% on sub-nanomolar gas samples, Brian also created a method to measure environmental levels of tritium ($^3H$) in water samples by the $^3He$ regrowth technique (Clarke et al., 1976), which has become the *de facto* state of the art in low level tritium measurements. In addition to being intrinsically simpler than the traditional low level method (which involved electrolytic enrichment combined with

low-level proportional gas counting), this method has proved to be more precise (by more than a factor of 4) and extended the detection limit to lower levels (by as much as an order of magnitude) (Bayer et al., 1989; Jenkins et al., 1983; Lott and Jenkins, 1998).

Brian was inventive and ingenious in the laboratory with a remarkable ability to recognize opportunities where no one else could, and to pursue them to ultimate success. He was an accomplished glass blower, and constructed vacuum lines and

research apparati from scratch using many different kinds of glass (see Figure 8). Brian kept unusual work hours when not teaching; generally arriving at the laboratory after lunch-time and toiling into the night. Working with him was a delight due to his good nature and whimsical sense of Irish humor, but sometimes challenging because he was an inveterate prankster.

### 4.2 H. Göte Östlund (1923-2016)

H. Göte Östlund was a pioneer in U.S. ocean tracer measurements, establishing a world class, low-level counting laboratory

dedicated to the measurement of tritium and radiocarbon at the University of Miami. He made distinguished contributions to Ocean, Atmosphere, and Groundwater sciences, in particular to understanding of the time scales of the transport of fluids through these systems. Östlund had a life-long devotion to the high-quality measurement of radioactive species. He received a B.S. in chemistry in 1949 and a Ph.D. in chemistry in 1958. Both were from the University of Stockholm. Between 1959



and 1961 Göte developed the electrolytic enrichment of tritium and deuterium for low-level environmental tritium measurements by gas proportional counting at the Radioactive Dating Laboratory of the Swedish Geological Survey. In the early 1960s, he came to the Rosenstiel School of the University of Miami. At the Rosenstiel School, he built a world-class tritium and radiocarbon counting laboratory that set new standards for low level counting. His laboratory also processed the
samples rapidly, and he generously shared his data with colleagues. As a result, relatively routine collection and analysis of large quantities of samples in a timely manner enabled oceanographers to use the tracer data to gain new insights into the times scales of oceanographic processes. His work paved the way for the acceptance of the next generation of tracer oceanographers, those measuring tritium and helium-3 by mass spectrometry, and those measuring the chlorofluorocarbons.

He played a key role in the creation and execution of early global ocean survey programs. Although his early interest
focused on atmospheric transport, it quickly extended to the hydrologic cycle and the oceans. He developed electrolytic enrichment techniques for low-level environmental tritium measurements by gas proportional counting (see Figure 9).

Östlund was a member of the scientific steering committee for the Geochemical Ocean Sections Study (GEOSECS), which was the first global scale survey of chemical, isotopic and radiochemical tracers in the ocean (1972-1978). He produced the first large-scale, high quality mapping of the distribution of tritium in the oceans, which opened oceanographer's eyes to the
dynamic and rapid penetration of bomb-produced tracers into the deep ocean.

Göte Östlund published over 100 papers in peer-reviewed journals on a wide range of subjects. Although his early interests focused on many areas including atmospheric transport, they quickly extended to the hydrologic cycle and the ocean where he applied radioactive tracers to a spectrum of scientific problems. As a student, he participation in the discovery of the anaesthetic Xylocain. Soon after coming to Miami, he used tritium to show that evaporation from the ocean is the major fuel
source for hurricanes. To collect samples he even flew into the eye of a hurricane. He and a colleague were the first to use tritium data to show that vertical mixing in the upper layers of the open ocean was an order of magnitude smaller than predicted by mass balance and theory. This was corroborated twenty years later by other investigators using new techniques. A major interest of his was the Arctic Ocean. There he quantified the contributions from ice melt, runoff and precipitation to the freshwater budget. This budget plays a critical role in the global overturning circulation, which is the leading candidate for
modulating decadal to centennial climate.

Göte was involved in the planning and implementation and served on the scientific steering committees of early global change programs: Geochemical Ocean Sections (GEOSECS), which was the first global scale survey of chemical, isotopic and radiochemical tracers in the ocean (1972-1978), followed in the 1980s by Transient Tracers in the Oceans (TTO). The data his laboratory produced, collected under the auspices of these programs, have furthered our understanding of the time scales of
ocean processes. For example, the data have been used to estimate the flux of anthropogenic carbon dioxide into the ocean, the rate of exchange between the atmosphere and ocean, and rates of deep-water formation. He produced the first large-scale, high quality mapping of the distribution of tritium in the ocean, which opened a new vista on the dynamic and rapid penetration of bomb-produced tracers into the deep western North Atlantic Ocean. Göte's leadership as a member and coordinator of the




scientific advisory committee for GEOSECS and TTO and his vision and credibility in seeing that an accelerator mass spectrometry facility for $^{14}$C analysis was established in the United States - had a large influence on ocean science. The big oceanographic programs of the past 50 years (GEOSECS through WOCE, CLIVAR and GO-SHIP) have provided platforms for obtaining large quantities of high quality tracer data.

Göte was soft-spoken and gentle in demeanor, and generous with his time, advice, and data. He set an example and benchmark for subsequent generations of tracer geochemists for responsibility, honesty, and fairness.

### 4.3 Financial Support

This synthesis work was funded under the auspices of a U.S. National Science Foundation grant number OCE-1434000.

Financial support for the actual measurements came from a wide variety of different research grants from many agencies in many countries, far too numerous to list here. The first author (Jenkins) is grateful to a number of U.S. funding sources, most notably the National Science Foundation, NOAA, DOE, and ONR.

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

**6 Tables**

**Table 1:** Fields (columns) in the main data table

| Field Name | Field Type | Field Description |
|---|---|---|
| ExpoCode | Short Text | Unique string identifying cruise/expedition |
| Sect_ID | Short Text | String identifying Ocean Section (WOCE/CLIVAR/GEOTRACES) |
| Station | Short Text | Station name or number |
| CastNo | Short Text | Cast name or number at that Station |
| Bottle | Short Text | Bottle name or number on that cast |
| StaDate | Number | Decimal year of sampling |
| Latitude | Number | North latitude in decimal degrees (from -90 to +90) |
| Longitude | Number | East longitude in decimal degrees (from -180 to +180) |
| StaDepth | Number | Bottom depth at station location in meters |
| Pressure | Number | Bottle depth (actually pressure) measured in dbar |
| Temperature | Number | In Situ temperature in degrees centigrade |
| Temperature_Flag | Integer | Temperature quality flag (see QF table) |
| Salinity | Number | Sample salinity in PSU |
| Salinity_Flag | Integer | Salinity quality flag (see QF table) |
| Oxygen | Number | Dissolved oxygen in umol/kg |





| Oxygen_Flag | Integer | Dissolved oxygen flag (see QF table) |
|---|---|---|
| Tritium | Number | Tritium in TU at time of sampling |
| Tritium_Error | Number | Uncertainty in TU at time of sampling |
| Tritium_Flag | Integer | Tritium quality flag (see QF table) |
| Tritium_PI | Short Text | Principle investigator or measurer of tritium |
| Tritium_PI_Inst | Short Text | Institution or laboratory where tritium was measured |
| Tritium_Method | Short Text | Short descriptor of tritium sampling/analysis method |
| DelHe3 | Number | Helium isotope ratio anomaly relative to atmosphere in percent |
| DelHe3_Error | Number | Uncertainty in helium isotope ratio anomaly |
| DelHe3_Flag | Integer | Helium isotope ratio anomaly quality flag (see QF table) |
| Helium | Number | Dissolved helium concentration in nmol/kg |
| Helium_Error | Number | Uncertainty in dissolved helium concentration in nmol/kg |
| Helium_Flag | Integer | Dissolved Helium quality flag (see QF table) |
| Neon | Number | Dissolved neon concentration in nmol/kg |
| Neon_Error | Number | Uncertainty in dissolved neon concentration in nmol/kg |
| Neon_Flag | Integer | Dissolved neon quality flag (see QF table) |
| Helium_PI | Short Text | Principle investigator or measurer of helium (and neon) |
| Heliulm_PI_Ins | Short Text | Institution or laboratory where tritium was measured |
| Helium_Method | Short Text | Short descriptor of helium sampling/analysis method |
| Reference_Code | Short Text | Data origin or link to paper discussing data |
| Reference_Source | Short Text | Data Source within reference (e.g., table, figure) if relevant |
| DOI | Short Text | Digital Object Identifer of orginal data set (if existing) |
| Comment | Short text | Additional information or comments |
| Record_ID | Long Integer | Unique record identifier number |

**Table 2:** Quality flag meaning

| Quality Flag Number | Meaning |
|---|---|
| 2 | Normal data, no problems reported |
| 3 | Questionable data: may not fit profile or some other doubt |
| 6 | Average of 2 or more measurements |
| 9 | Missing (null) data |



**Table 3:** Contributing analysts that are not authors on this paper

| A.E. Bainbridge | UCSD, La Jolla, CA, USA |
|---|---|
| R. Bayer | U. Heidelberg, Heidelberg, Germany |
| F. Begemann | U. Chicago, Chicago, IL, USA |
| U. Beyerle | ETH, Zurich, Switzerland |
| W.S. Broecker | LDEO, Pallisades, NY, USA |
| M. Butzin | University of Bremen, Bremen, Germany |
| W.B. Clarke* | McMaster University, Hamilton, ON, Canada |
| K.O. Dockins | UCSD, La Jolla, CA, USA |
| H.G. Dorsey | RSMAS, Miami, FL, USA |
| E. Eriksson | IMS, Stockholm, Sweden |
| E. Fourré | CEA-Saclay, France |
| B.J. Giletti | LDEO, Pallisades, NY, USA |
| A.V. Grosse | RITU, Philadelphia, PA, USA |
| J.R. Harries | Australian AEC, Sutherland, NSW, Australia |
| T, Kaji | Kyushu University, Fukuoka, Japan |
| S. Kaufman | U. Chicago, Chicago, IL, USA |
| J.L. Kulp | LDEO, Pallisades, NY, USA |
| W.F. Libby | U. Chicago, Chicago, IL, USA |
| D.E. Lott III | WHOI, Woods Hole, MA, USA |
| A.Luden | LDEO, Pallisades, NY, USA |
| L. Merlivat | Sorbonne University, Paris, France |
| R. Michel | UCSD, La Jolla, CA, USA |
| Y. Miyake | GRA, Tokyo, Japan |
| K.O. Munnich | U. Heidelberg, Heidelberg, Germany |
| A.O. Nier | U. Minnesota, Minneapolis, MN, USA |
| M. Nonaka | IPRC & SOES, Tokyo, Japan |
| H.G. Ostlund* | RSMAS, Miami, FL, USA |
| C. Postlethwaite | NOC-SOES, Southampton, U.K. |
| P.D. Quay | University of Washington, Seattle, WA, USA |
| R.S.H.R. Stanley | WHOI, Woods Hole, MA, USA |



| | |
|---|---|
| S. Stark | NOC-SOES, Southampton, U.K. |
| R. Steinfeldt | IUP, University Bremen, Germany |
| H.E. Suess | UCSD, La Jolla, CA, USA |
| J. Sűltenfuss | IUP, University Bremen, Germany |
| N. Takahata | ORI, University of Tokyo, Tokyo, Japan |
| A.Tamuly | University of Quebec, Rimouski, PQ, Canada |
| C.B. Taylor | INS, Lower Hutt, New Zealand |
| Z. Top | RSMAS, Miami, FL, USA |
| T. Torgersen | WHOI, Woods Hole, MA, USA |
| K.A. Van Scoy | RSMAS, Miami, FL, USA |
| C. Walker | WHOI, Woods Hole, MA, USA |
| W. Weiss | U. Heidelberg, Heidelberg, Germany |
| P.M. Williams | UCSD, La Jolla, CA, USA |

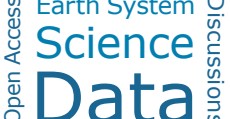



**7 Figures**

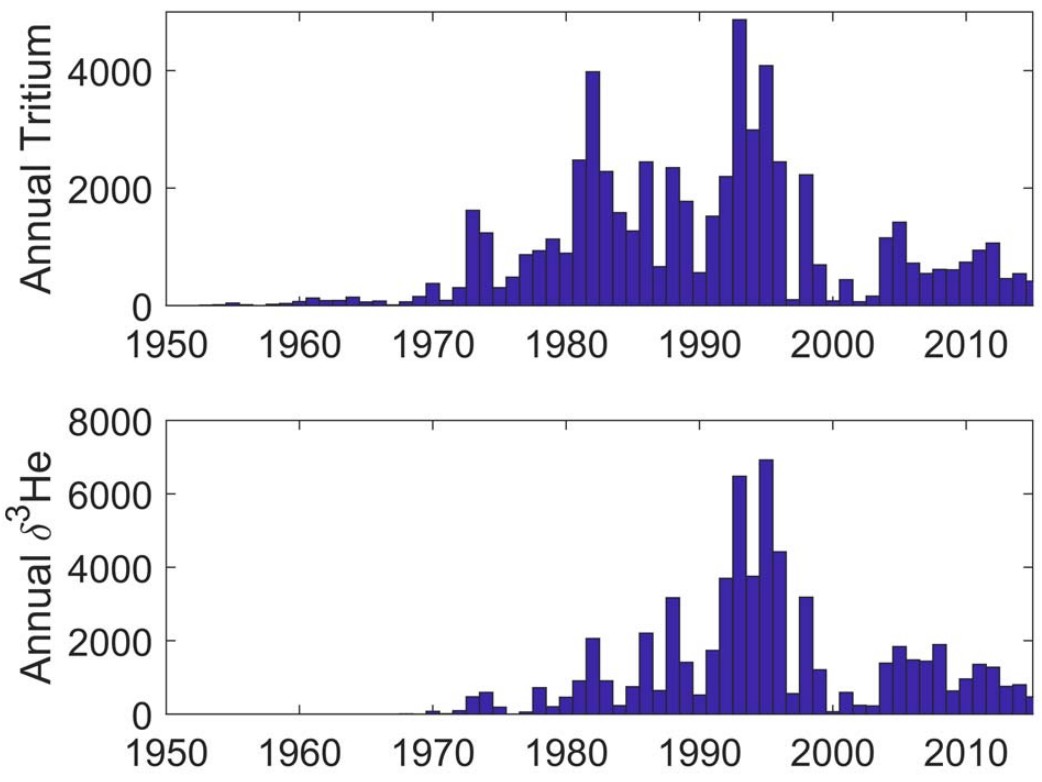

**Figure 1: Time distributions of annual tritium (upper panel) and helium (lower panel) measurements.**

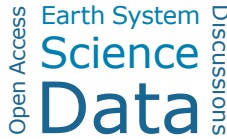

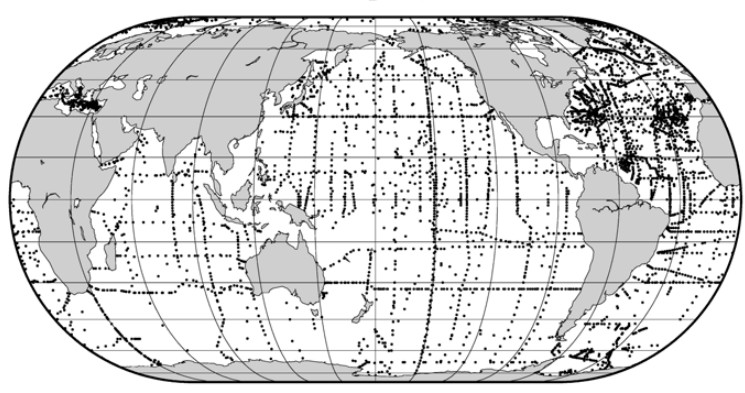

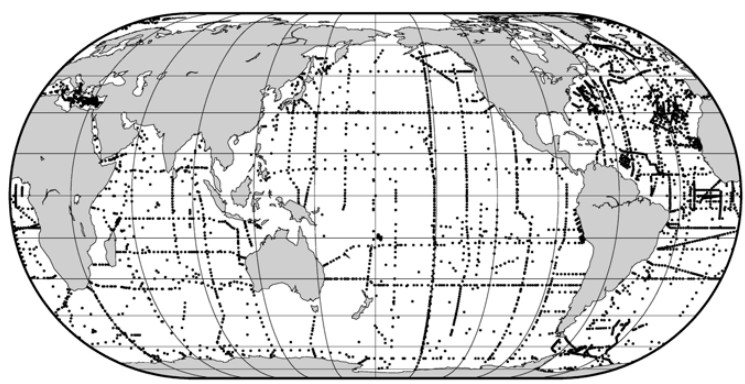

**Figure 2: Tritium and helium sample locations**



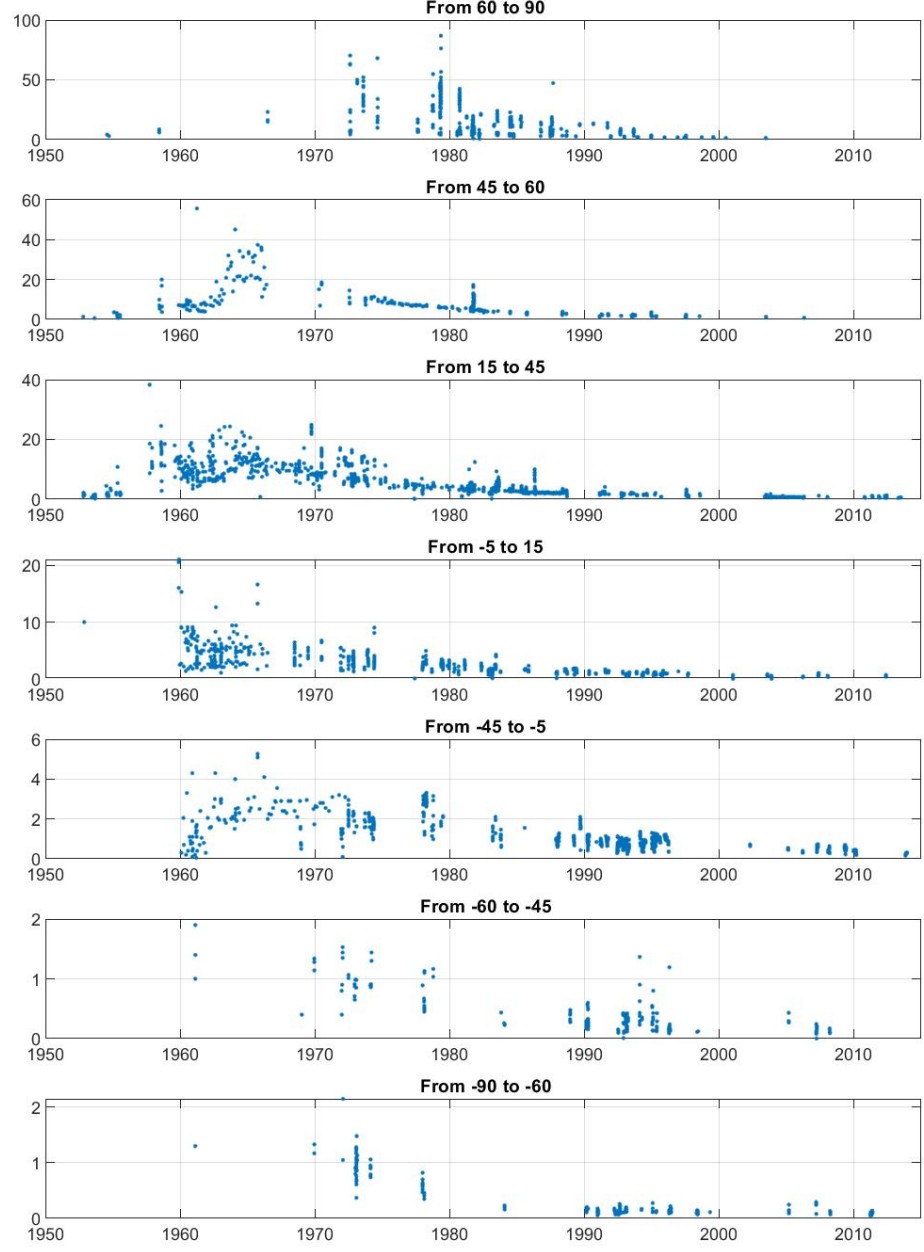

**Figure 3: Global ocean surface water (depth < 50 m) tritium concentrations (in TU) for selected latitude bands.**





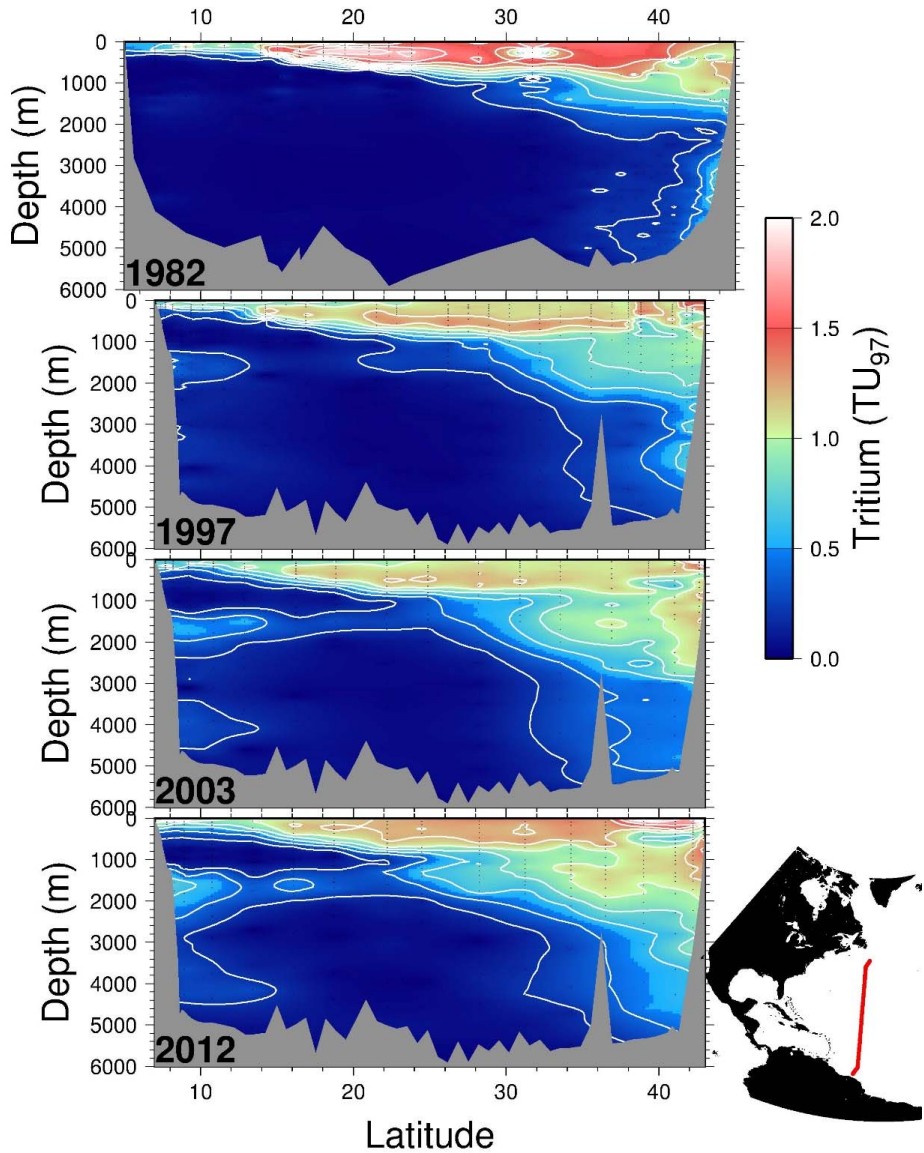

**Figure 4: Four meridional tritium sections along roughly 52°W in the North Atlantic taken in 1982, 1997, 2003, and 2012. The tritium concentrations have be decay corrected to a common time (January 1, 1997) for comparison. Contour intervals are 0.2 TU$_{97}$ and measurement uncertainties are of order 0.01 TU$_{97}$ or better. Due to differences in cruise tracks, the topography for the 1982 occupation differs from the others.**



**Figure 5: Four meridional δ³He sections along roughly 52°W in the North Atlantic taken in 1982, 1997, 2003, and 2012. Contour intervals are 1% and measurement uncertainties are 0.15%.**





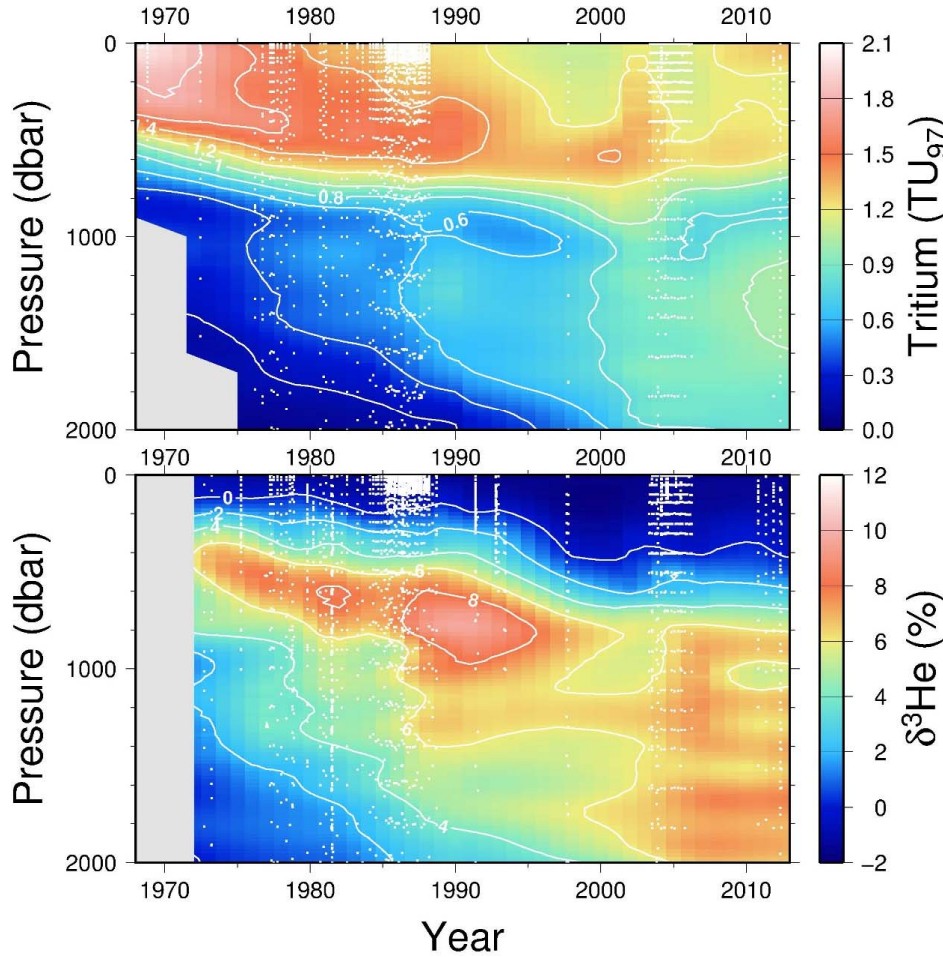

**Figure 6: A time series of tritium (upper) and helium isotope measurements in the vicinity of Bermuda (North Atlantic). Tritium values have been decay-corrected to a common time (January 1, 1997). White dots indicate sampling depths and times.**



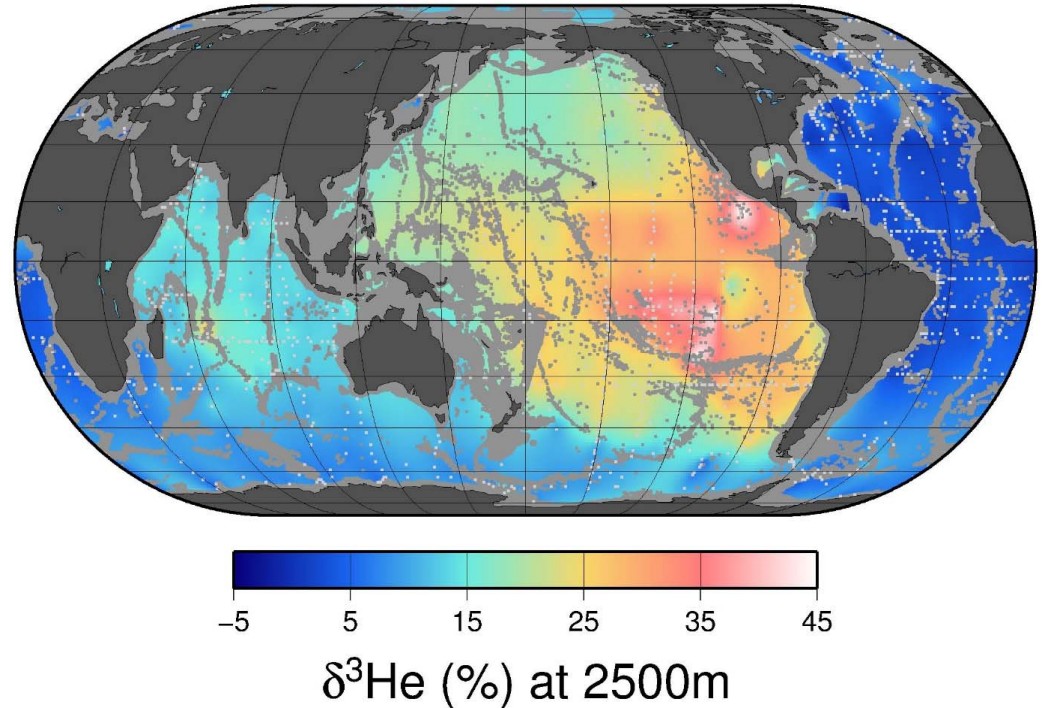

**Figure 7: A map of δ³He values at approximately 2500 m depth. The values plotted are simply an average of all measurements within a 1° square between 2250 and 2750 dbar. Depths shallower than 2500 m are masked in gray and sampling locations indicated by light gray dots. (Perhaps Reiner or I may do something better here, such as interpolate onto a neutral density surface.)**



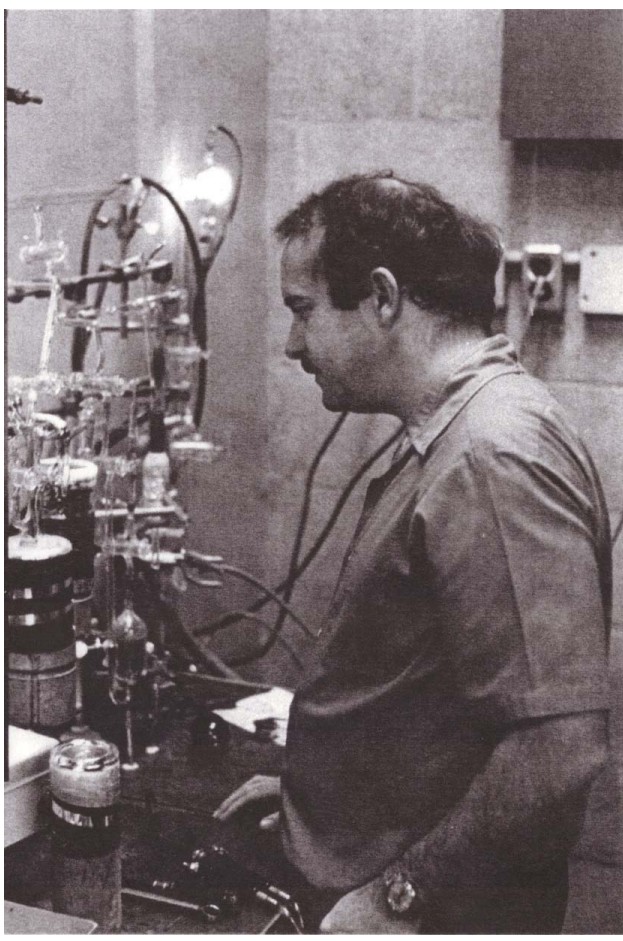

**Figure 8: W. Brian Clarke, working on a high vacuum helium extraction apparatus (early 1970s)**




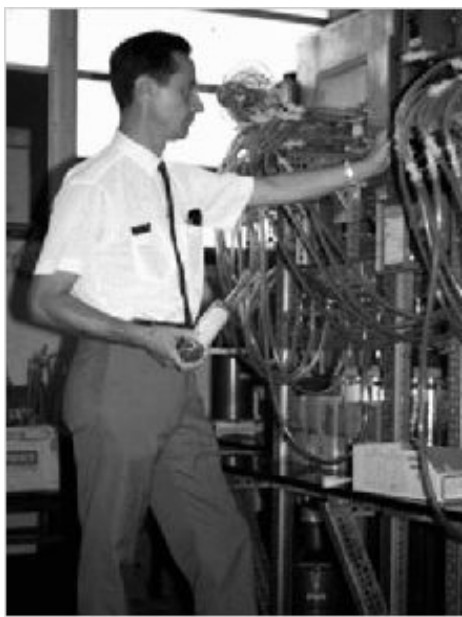

**Figure 9: H. Göte Östlund preparing a gas sample for low level counting analysis (mid 1960s).**