# Peer review of "A comprehensive global oceanic dataset of helium isotope and tritium measurements"

_Earth System Science Data, 2018_

## Referee Comment (RC1) · Anonymous Referee #1 · 11 Dec 2018

The authors have organized all existing oceanic measurements of helium-isotope and tritium data into a single comprehensive data set, complete with uncertainty estimates and valuable metadata on data quality and measurement methods. This tracer data is invaluable to oceanography and collecting it all in a single place is a significant service to the scientific community. This short paper accompanying the data set is appropriate and useful. I found the descriptions of the analytical methods useful and I enjoyed the tributes to pioneers Clarke and Ostlund. Subject to minor revisions, I recommend publication.

My main comment is that with 22% of the data being neon measurements, there should be at least a paragraph discussing the use of the neon data with some references, and perhaps a figure on neon. I assume neon, having only stable natural isotopes, is used

to constrain air-sea gas exchange, but I don't know of all its uses in oceanography and suspect I'm not alone in this. Some discussion on neon would be useful. Perhaps a graphic on observed neon surface saturation could be shown.

Minor comments just concern occasional clarity issues and typographical/grammatical errors:

P1 L19 typo: . . . with a half-life of 12.3 years . . . "half-life of" is missing

P2 L24 typo: either measurement "of" or "measurement" should be "measuring"

P2 L27 water samples -> gas samples ? (electrolysis enriches the gas, not the water)

P3 L13 "magnetic sector instrument" please explain what this means on first usage (e.g., is it different from a mass spectrometer deflecting ions as they traverse a magnetic field?)

P3 L22: could the ionization state on 4He and 20Ne in the iron currents be indicated here?

P3 L22 "helium/neon" -> "helium or neon" otherwise this is suggesting that the helium/neon ratio is somehow involved, which presumably is not the intent.

P4 L2 typo: additional -> addition

P4 L13: "The dataset . . .." On a first reading, I immediately wanted to know what the file format was – it might be nice to add a parenthetical "(digital formats are specified below)".

P4 L14 "in principle" seems inappropriate here. Perhaps you mean: "Where available, this provides . . ."

P4 L15 "methods fields" – delete "methods"

P4 L18 typo: "in the Table 1" -> "in Table 1"

Section 2.5 (Data formats and availability) Why is data in netcdf format not included?

Arguably this is the standard for portable data in the geosciences.

Section 3 "Graphics and examples" seems like a very poor heading for a section because it is so generic and non-descriptive. How about "Scope and nature of the dataset" instead?

P5 L4 "... for maximum flexibility. For maximum flexibility, we ..." Delete the first "for maximum flexibility"

P5 L10 "example graphics" delete "example"

P5 L22 delete "basically" or make precise qualifying statement

P6 L5 There were also a number atmospheric, as well as submarine, nuclear tests in the Southern Hemisphere by the British and French militaries. Please add a few sentences of discussion on this and perhaps point out if this is discernible in the available oceanographic data (some of the spikes in the southern bands?) What was the estimated tritium production by nuclear testing in each hemisphere?

L11 P6 there is a grammar issue: "A benchmark observations" should this be singular? – please check sentence and fix.

L13 P6 Figure 5 should be Figure 4.

L14 P6 "along a section along" -> "along a section at"

L20 P6 "Equally important is the bottom-contour-hugging ..." This sentence is very unclear. What does it mean for a "level" to hug a "bottom contour"? It is unclear what in the figure is being described – please reword for clarity.

L23 P6 Figure 4 should be Figure 5. Also "those" sections – obscure – same as in Figure 4 – please restate.

L30 P6 "We also include a ..." You need to explicitly reference "Figure 6"

L1 P7 "excess 3He" what does "excess" mean here? Do you mean elevated relative to

the atmospheric 3He/4He ratio or excess over global mean or something else? Please clarify and perhaps just refer to delta3He as that is precisely defined.

P9 L9-10 and L16-17 "Although his early . . . and the ocean" This is a repeat of the exact same sentence – should probably have been deleted from L9-10

P9 L19 "fuel" seems a bit like the wrong word here. "Energy source" ?

Figure 4: why is the decay-corrected tritium in the upper ocean higher in 2012 than in 2003? This seems like a prominent feature of this figure that should be briefly discussed in the text.

Figure 6: "vicinity" what is the lat and lon?

Figure 7: "Perhaps Reiner and I . . ." Please remove internal commentary among co-authors (i.e., please proof-read your paper and captions . . .)

―――――――――――――――――

---

## Author Comment (AC1) · 2 Jan 2019

Due to an unfortunate oversight, we neglected to include R. Newton (LDEO, Columbia University, U.S.A.) as a co-author on this manuscript. This will be rectified in the revised version. Please consider him as one of the authors.
* * *

---

## Referee Comment (RC2) · Anonymous Referee #2 · 3 Jan 2019

Jenkins et al. (ESSD-2018-136) introduces a dataset of oceanic helium and tritium measurements, which is open to the world via www with DOI. The dataset is definitely valuable for any earth scientists, especially oceanographers. Analytical methods used are valid. Various formats of the dataset provided are helpful. I have nothing to say without some tiny/minor points before its publication on ESSD.

P1L15: Add "." between depth and When

P1L19: with a half-life of 12.3y?

P2L2-5: Numerous refs are nice but their generations seem biased in 20th century. Please introduce nice papers in 21st century if possible.

P2L12-14: In turn, generations of refs here seem biased in 21st century. Please intro-

duce classic/pioneering papers if possible.

P2L16: Please add refs for each of these programs.

P3L17: How deep and old? Please introduce approximate criteria.

P3L23: 4He and 20Ne are NOT ion currents.

P4L6: Please add density into the dataset because of its importance in oceanography.

P5L2: I request the dataset distributed by another (non-US) web server for more robust accessibility if possible.

P6L13: figure number?

P6L23: figure number?

P6L30: Figure 6?

P7L5: How large and small?

P7L18: I cannot understand why numerous hard workers in Table 3 are not included in the authorship list. I know analyses of helium and tritium in seawater (also sampling of oceanic water) are definitely hard. I think the list should include the hard workers.

Figure 7: Another map for 3000m (or 3500m) seems nice to figure out the impact from hydrothermal activity on the ridges represented in the current 2500m map.

---

## Author Response (AR1)

**Response to reviewers for "A comprehensive global oceanic dataset of helium isotope and tritium measurements" by William J. Jenkins, Scott C. Doney , Michaela Fendrock, Rana Fine,  Toshitaka Gamo, Philippe Jean-Baptiste, Robert Key, Birgit Klein, John E. Lupton1, Robert Newton, Monika Rhein, Wolfgang Roether, Yuji Sano, Reiner Schlitzer, Peter Schlosser, Jim Swift**

*Anonymous Referee #1*
*The authors have organized all existing oceanic measurements of helium-isotope and tritium data into a single comprehensive data set, complete with uncertainty estimates and valuable metadata on data quality and measurement methods. This tracer data is invaluable to oceanography and collecting it all in a single place is a significant service to the scientific community. This short paper accompanying the data set is appropriate and useful. I found the descriptions of the analytical methods useful and I enjoyed the tributes to pioneers Clarke and Ostlund. Subject to minor revisions, I recommend publication.*

Thank you for the kind comments.

*My main comment is that with 22% of the data being neon measurements, there should be at least a paragraph discussing the use of the neon data with some references, and perhaps a figure on neon. I assume neon, having only stable natural isotopes, is used to constrain air-sea gas exchange, but I don't know of all its uses in oceanography and suspect I'm not alone in this. Some discussion on neon would be useful. Perhaps a graphic on observed neon surface saturation could be shown.*

The Reviewer makes a very good point, we had overlooked spelling out the relative importance of the He and Ne concentration measurements. We added a paragraph in section 2.3 explaining the value of both measurements, and presented a composite scatter plot of their respective supersaturations in global surface waters.

*Minor comments just concern occasional clarity issues and typographical/grammatical errors:*

*P1 L19 typo: : : : with a half-life of 12.3 years : : : "half-life of" is missing*

Thank you for catching that. This has been corrected.

*P2 L24 typo: either measurement "of" or "measurement" should be "measuring"*

Thank you for catching that. This has been corrected.

*P2 L27 water samples -> gas samples ? (electrolysis enriches the gas, not the water)*

Thank you for catching that. This has been corrected.

*P3 L13 "magnetic sector instrument" please explain what this means on first usage (e.g., is it different from a mass spectrometer deflecting ions as they traverse a magnetic field?)*

That is correct. We have added a brief phrase to make it clearer.

*P3 L22: could the ionization state on 4He and 20Ne in the iron currents be indicated here?*

This has now been done.

*P3 L22 "helium/neon" -> "helium or neon" otherwise this is suggesting that the helium/neon ratio is somehow involved, which presumably is not the intent.*

Good point. Now corrected.

*P4 L2 typo: additional -> addition*

Now corrected.

*P4 L13: "The dataset : : :." On a first reading, I immediately wanted to know what the file format was – it might be nice to add a parenthetical "(digital formats are specified below)".*

Phrase added.

*P4 L14 "in principle" seems inappropriate here. Perhaps you mean: "Where available, this provides : : :"*

Rephrased.

*P4 L15 "methods fields" – delete "methods"*

Deleted

*P4 L18 typo: "in the Table 1" -> "in Table 1"*

Corrected.

*Section 2.5 (Data formats and availability) Why is data in netcdf format not included? Arguably this is the standard for portable data in the geosciences.*

We now provide a netcdf version of the data.

*Section 3 "Graphics and examples" seems like a very poor heading for a section because it is so generic and non-descriptive. How about "Scope and nature of the dataset" instead?*

This is a good suggestion. We have changed the section heading accordingly.

*P5 L4 ": : : for maximum flexibility. For maximum flexibility, we : : :" Delete the first "for maximum flexibility"*

Phrase deleted.

*P5 L10 "example graphics" delete "example"*

Deleted.

*P5 L22 delete "basically" or make precise qualifying statement*

Word deleted.

*P6 L5 There were also a number atmospheric, as well as submarine, nuclear tests in the Southern Hemisphere by the British and French militaries. Please add a few sentences of discussion on this and perhaps point out if this is discernible in the available oceanographic data (some of the spikes in the southern bands?) What was the estimated tritium production by nuclear testing in each hemisphere?*

We added some additional discussion, but please understand that we have tried to strike a reasonable balance by describing the general features of the data set without digressing into too much detail regarding the causal forces behind it. The latter would be the subject of a much larger study. We also urge some caution regarding the interpretation of "spikes" in the latitude bands, particularly early in the records, because they may well reflect local (anomalous) fallout extremes rather than excursions in regional inventories. They are, after all, individual measurements.

*L11 P6 there is a grammar issue: "A benchmark observations" should this be singular? – please check sentence and fix.*

This has been rectified.

*L13 P6 Figure 5 should be Figure 4.*

Figure numbering has been updated.

*L14 P6 "along a section along" -> "along a section at"*

Redundancy fixed.

*L20 P6 "Equally important is the bottom-contour-hugging : : :" This sentence is very unclear. What does it mean for a "level" to hug a "bottom contour"? It is unclear what in the figure is being described – please reword for clarity.*

This has been re-worded to be clearer.

*L23 P6 Figure 4 should be Figure 5.*

Figure numbers corrected.

*Also "those" sections – obscure – same as in Figure 4 – please restate.*

Done.

*L30 P6 "We also include a : : :" You need to explicitly reference "Figure 6"*

Done.

We agree. We now revert to the helium isotope ratio anomaly, for which there is a clear definition.

*P9 L9-10 and L16-17 "Although his early : : : and the ocean" This is a repeat of the exact same sentence – should probably have been deleted from L9-10*

Redundant sentence deleted.

*P9 L19 "fuel" seems a bit like the wrong word here. "Energy source" ?*

Thank you, "energy source" is a better term.

*Figure 4: why is the decay-corrected tritium in the upper ocean higher in 2012 than in 2003? This seems like a prominent feature of this figure that should be briefly discussed in the text.*

We have added a brief explanation.

*Figure 6: "vicinity" what is the lat and lon?*

Added.

*Figure 7: "Perhaps Reiner and I : : :" Please remove internal commentary among coauthors (i.e., please proof-read your paper and captions : : :)*

Sorry. Comment removed.

*Anonymous Referee #2*
*Jenkins et al. (ESSD-2018-136) introduces a dataset of oceanic helium and tritium measurements, which is open to the world via www with DOI. The dataset is definitely valuable for any earth scientists, especially oceanographers. Analytical methods used are valid. Various formats of the dataset provided are helpful. I have nothing to say without some tiny/minor points before its publication on ESSD.*

*P1L15: Add "." between depth and When*

Period added.

*P1L19: with a half-life of 12.3y?*

Corrected.

*P2L2-5: Numerous refs are nice but their generations seem biased in 20th century. Please introduce nice papers in 21st century if possible.*

These references are largely weighted to the 20$^{th}$ century because the work involved was largely published there. We don't want to create an extensive bibliography of all work done in this area as it would detract from the main purpose of the paper.

*P2L12-14: In turn, generations of refs here seem biased in 21st century. Please introduce classic/pioneering papers if possible.*

We included only the most recent determinations of the global 3He flux because they are by far the most accurate. Those papers refer to the earlier, foundational, but less accurate determinations.

*P2L16: Please add refs for each of these programs.*

We have added the references.

*P3L17: How deep and old? Please introduce approximate criteria.*

The second part of the sentence actually provides the criteria. Rather than specify a specific depth, which would not apply in some areas than others, we give an example.

*P3L23: 4He and 20Ne are NOT ion currents.*

We have corrected this (see Reviewer 1 comments).

*P4L6: Please add density into the dataset because of its importance in oceanography.*

Density is a derived and calculable property given the sample's pressure, temperature, and salinity. In addition, it requires a reference pressure (i.e., does one refer to the sea surface, some fixed level – e.g., 3000 dbar – or do you use neutral density?).

*P5L2: I request the dataset distributed by another (non-US) web server for more robust accessibility if possible.*

We have now supplied an EU-based mirror site (referred to in the manuscript).

*P6L13: figure number?*

Corrected.

*P6L23: figure number?*

Corrected.

*P6L30: Figure 6?*

Corrected.

*P7L5: How large and small?*

Descriptors added.

*P7L18: I cannot understand why numerous hard workers in Table 3 are not included in the authorship list. I know analyses of helium and tritium in seawater (also sampling of oceanic water) are definitely hard. I think the list should include the hard workers.*

The reviewers request a formidable task: we are well aware that the hard work and efforts made by many people in making these measurements. The authorship of this article reflects the contributions of the individuals assembling this dataset, not the original generation of the data. We have made considerable effort to provide within the dataset the correct attribution for the data originators and we hope this adequate recognition of their contributions.

*Figure 7: Another map for 3000m (or 3500m) seems nice to figure out the impact from hydrothermal activity on the ridges represented in the current 2500m map.*

We have included a contrasting map at the 4000m horizon as well, which we think better emphasizes the differences.

[revised manuscript text omitted]

*Anonymous Referee #1*

*The authors have organized all existing oceanic measurements of helium-isotope and tritium data into a single comprehensive data set, complete with uncertainty estimates and valuable metadata on data quality and measurement methods. This tracer data is invaluable to oceanography and collecting it all in a single place is a significant service to the scientific community. This short paper accompanying the data set is appropriate and useful. I found the descriptions of the analytical methods useful and I enjoyed the tributes to pioneers Clarke and Ostlund. Subject to minor revisions, I recommend publication.*

Thank you for the kind comments.

*My main comment is that with 22% of the data being neon measurements, there should be at least a paragraph discussing the use of the neon data with some references, and perhaps a figure on neon. I assume neon, having only stable natural isotopes, is used to constrain air-sea gas exchange, but I don't know of all its uses in oceanography and suspect I'm not alone in this. Some discussion on neon would be useful. Perhaps a graphic on observed neon surface saturation could be shown.*

The Reviewer makes a very good point, we had overlooked spelling out the relative importance of the He and Ne concentration measurements. We added a paragraph in section 2.3 explaining the value of both measurements, and presented a composite scatter plot of their respective supersaturations in global surface waters.

*Minor comments just concern occasional clarity issues and typographical/grammatical errors:*

*P1 L19 typo: : : : with a half-life of 12.3 years : : : "half-life of" is missing*

Thank you for catching that. This has been corrected.

*P2 L24 typo: either measurement "of" or "measurement" should be "measuring"*

Thank you for catching that. This has been corrected.

*P2 L27 water samples -> gas samples ? (electrolysis enriches the gas, not the water)*

Thank you for catching that. This has been corrected.

*P3 L13 "magnetic sector instrument" please explain what this means on first usage*
*(e.g., is it different from a mass spectrometer deflecting ions as they traverse a magnetic field?)*

That is correct. We have added a brief phrase to make it clearer.

*P3 L22: could the ionization state on 4He and 20Ne in the iron currents be indicated here?*

This has now been done.

*P3 L22 "helium/neon" -> "helium or neon" otherwise this is suggesting that the helium/neon ratio is somehow involved, which presumably is not the intent.*

Good point. Now corrected.

*P4 L2 typo: additional -> addition*

Now corrected.

*P4 L13: "The dataset : : :." On a first reading, I immediately wanted to know what the file format was – it might be nice to add a parenthetical "(digital formats are specified below)".*

Phrase added.

*P4 L14 "in principle" seems inappropriate here. Perhaps you mean: "Where available, this provides : : :"*

Rephrased.

*P4 L15 "methods fields" – delete "methods"*

Deleted

*P4 L18 typo: "in the Table 1" -> "in Table 1"*

Corrected.

*Section 2.5 (Data formats and availability) Why is data in netcdf format not included? Arguably this is the standard for portable data in the geosciences.*

We now provide a netcdf version of the data.

*Section 3 "Graphics and examples" seems like a very poor heading for a section because it is so generic and non-descriptive. How about "Scope and nature of the dataset" instead?*

This is a good suggestion. We have changed the section heading accordingly.

*P5 L4 ": : : for maximum flexibility. For maximum flexibility, we : : :" Delete the first "for maximum flexibility"*

Phrase deleted.

*P5 L10 "example graphics" delete "example"*

Deleted.

*P5 L22 delete "basically" or make precise qualifying statement*

Word deleted.

5 *P6 L5 There were also a number atmospheric, as well as submarine, nuclear tests in the Southern Hemisphere by the British and French militaries. Please add a few sentences of discussion on this and perhaps point out if this is discernible in the available oceanographic data (some of the spikes in the southern bands?) What was the estimated tritium production by nuclear testing in each hemisphere?*

10 We added some additional discussion, but please understand that we have tried to strike a reasonable balance by describing the general features of the data set without digressing into too much detail regarding the causal forces behind it. The latter would be the subject of a much larger study. We also urge some caution regarding the interpretation of "spikes" in the latitude bands, particularly early in the records, because they may well reflect local (anomalous) fallout extremes rather than excursions in regional inventories. They are, after all, individual 15 measurements.

*L11 P6 there is a grammar issue: "A benchmark observations" should this be singular? – please check sentence and fix.*

20 This has been rectified.

*L13 P6 Figure 5 should be Figure 4.*

Figure numbering has been updated.
25
*L14 P6 "along a section along" -> "along a section at"*

Redundancy fixed.

30 *L20 P6 "Equally important is the bottom-contour-hugging : : :" This sentence is very unclear. What does it mean for a "level" to hug a "bottom contour"? It is unclear what in the figure is being described – please reword for clarity.*

This has been re-worded to be clearer.

35 *L23 P6 Figure 4 should be Figure 5.*

Figure numbers corrected.

*Also "those" sections – obscure – same as in Figure 4 – please restate.*
40
Done.

*L30 P6 "We also include a : : :" You need to explicitly reference "Figure 6"*

45 Done.

*L1 P7 "excess 3He" what does "excess" mean here? Do you mean elevated relative to the atmospheric 3He/4He ratio or excess over global mean or something else? Please clarify and perhaps just refer to delta3He as that is precisely defined.*
50

We agree. We now revert to the helium isotope ratio anomaly, for which there is a clear definition.

*P9 L9-10 and L16-17 "Although his early : : : and the ocean" This is a repeat of the exact same sentence – should probably have been deleted from L9-10*

Redundant sentence deleted.

*P9 L19 "fuel" seems a bit like the wrong word here. "Energy source" ?*

Thank you, "energy source" is a better term.

*Figure 4: why is the decay-corrected tritium in the upper ocean higher in 2012 than in 2003? This seems like a prominent feature of this figure that should be briefly discussed in the text.*

We have added a brief explanation.

*Figure 6: "vicinity" what is the lat and lon?*

Added.

*Figure 7: "Perhaps Reiner and I : : :" Please remove internal commentary among coauthors (i.e., please proof-read your paper and captions : : :)*

Sorry. Comment removed.

**Anonymous Referee #2**

*Jenkins et al. (ESSD-2018-136) introduces a dataset of oceanic helium and tritium measurements, which is open to the world via www with DOI. The dataset is definitely valuable for any earth scientists, especially oceanographers. Analytical methods used are valid. Various formats of the dataset provided are helpful. I have nothing to say without some tiny/minor points before its publication on ESSD.*

*P1L15: Add "." between depth and When*

Period added.

*P1L19: with a half-life of 12.3y?*

Corrected.

*P2L2-5: Numerous refs are nice but their generations seem biased in 20th century. Please introduce nice papers in 21st century if possible.*

These references are largely weighted to the 20$^{th}$ century because the work involved was largely published there. We don't want to create an extensive bibliography of all work done in this area as it would detract from the main purpose of the paper.

*P2L12-14: In turn, generations of refs here seem biased in 21st century. Please introduce classic/pioneering papers if possible.*

We included only the most recent determinations of the global 3He flux because they are by far the most accurate. Those papers refer to the earlier, foundational, but less accurate determinations.

*P2L16: Please add refs for each of these programs.*

We have added the references.

*P3L17: How deep and old? Please introduce approximate criteria.*

The second part of the sentence actually provides the criteria. Rather than specify a specific depth, which would not apply in some areas than others, we give an example.

*P3L23: 4He and 20Ne are NOT ion currents.*

We have corrected this (see Reviewer 1 comments).

*P4L6: Please add density into the dataset because of its importance in oceanography.*

Density is a derived and calculable property given the sample's pressure, temperature, and salinity. In addition, it requires a reference pressure (i.e., does one refer to the sea surface, some fixed level – e.g., 3000 dbar – or do you use neutral density?).

*P5L2: I request the dataset distributed by another (non-US) web server for more robust accessibility if possible.*

We have now supplied an EU-based mirror site (referred to in the manuscript) at http://odv.awi.de/data/ocean/jenkins-tritium-helium-data-compilation/

*P6L13: figure number?*

Corrected.

*P6L23: figure number?*

Corrected.

10  *P6L30: Figure 6?*

Corrected.

*P7L5: How large and small?*

Descriptors added.

*P7L18: I cannot understand why numerous hard workers in Table 3 are not included in the authorship list. I know analyses of helium and tritium in seawater (also sampling of oceanic water) are definitely hard. I think the list should*
20  *include the hard workers.*

The reviewers request a formidable task: we are well aware that the hard work and efforts made by many people in making these measurements. The authorship of this article reflects the contributions of the individuals assembling this dataset, not the original generation of the data. We have made considerable effort to provide within the dataset
25  the correct attribution for the data originators and we hope this adequate recognition of their contributions.

*Figure 7: Another map for 3000m (or 3500m) seems nice to figure out the impact from hydrothermal activity on the ridges represented in the current 2500m map.*

30  We have included a contrasting map at the 4000m horizon as well, which we think better emphasizes the differences.